# Assessment of Therapeutic Potential of a Dual AAV Approach for Duchenne Muscular Dystrophy

**DOI:** 10.3390/ijms241411421

**Published:** 2023-07-13

**Authors:** Sonia Albini, Laura Palmieri, Auriane Dubois, Nathalie Bourg, William Lostal, Isabelle Richard

**Affiliations:** 1Genethon, 91100 Evry, France; 2Université Paris-Saclay, Univ Evry, Inserm, Genethon, Integrare Research Unit UMR_S951, 91000 Evry, France

**Keywords:** gene therapy, AAV, dual vector, DMD, dystrophin, homologous recombination, concatemerization

## Abstract

Duchenne muscular dystrophy (DMD) is a yet incurable rare genetic disease that affects the skeletal and cardiac muscles, leading to progressive muscle wasting and premature death. DMD is caused by the lack of dystrophin, a muscle protein essential for the biochemical support and integrity of muscle fibers. Gene replacement strategies for Duchenne muscular dystrophy (DMD) employing the adeno-associated virus (AAV) face the challenge imposed by the limited packaging capacity of AAV, only allowing the accommodation of a short version of dystrophin (µDys) that is still far removed from correcting human disease. The need to develop strategies leading to the expression of a best performing dystrophin variant led to only few studies reporting on the use of dual vectors, but none reported on a method to assess in vivo transgene reconstitution efficiency, the degree of which directly affects the use of safe AAV dosing. We report here on the generation of a dual AAV vector approach for the expression of a larger dystrophin version (quasidystrophin) based on homologous recombination, and the development of a methodology employing a strategic droplet digital PCR design, to determine the recombination efficiency as well as the occurrence of unwanted concatemerization events or aberrant expression from the single vectors. We demonstrated that, upon systemic delivery in the dystrophic D2.B10-Dmd^mdx^/J (DBA2mdx) mice, our dual AAV approach led to high transgene reconstitution efficiency and negligible Inverted Terminal Repeats (ITR)-dependent concatemerization, with consequent remarkable protein restoration in muscles and improvement of muscle pathology. This evidence supports the suitability of our system for gene therapy application and the potential of this methodology to assess and improve the feasibility for therapeutic translation of multiple vector approaches.

## 1. Introduction

Duchenne muscular dystrophy (DMD) is a rare, X-linked recessive muscle disorder caused by mutations in the dystrophin gene and affects roughly 1/5000 male births [1,2,3]. The severity of DMD relies on the rapid progression of the disease characterized by muscle degeneration, fibro-adipose tissue replacement, and general muscle wasting, that leads to premature death from cardiac and respiratory complications. The different types of mutations in the *DMD* gene result in aberrant or absent production of the dystrophin protein [2]. Its main function is to protect the myofiber from mechanical stress due to muscle contraction by linking the intracellular cytoskeleton to the ECM via a multiprotein complex, the dystrophin-glycoprotein-complex (DGC). Besides its structural role, more and more evidence shows that dystrophin is also a scaffold for signaling proteins involved in processes regulating muscle function [4,5]. 

DMD being a monogenic disease, gene replacement therapy is one of the most promising approaches for its treatment. However, the open reading frame encoding dystrophin (*DMD*) is >11 kb long, a length that is far beyond the capacity of adeno-associated vectors (AAVs), the currently most frequently employed vector for gene transfer in the muscle. The initial discovery that truncated dystrophins are still functional [6] provided the rationale to use smaller versions of dystrophin, fitting with AAVs capacity, that are currently being investigated in clinical trials. However, gene therapy employing the microdystrophin gene (µDys) can only attenuate the disease that evolves in the milder, yet invaliding, Becker muscular dystrophy. This limitation has thus inspired several projects to try to generate a larger dystrophin version that is closer to the native protein using more than one AAV vector and exploiting the innate ability of AAV genomes to undergo intermolecular concatemerization [7,8,9,10,11]. In such strategies, the reconstitution of the full transgene can be generated by homologous recombination (HR) of the two AAV vectors carrying homologous overlapping sequences or ITR-mediated head-to-tail concatemerization and trans-splicing that produces mature mRNA [11,12,13]. It is notable that over recent years, the efficiency and safety of the overlapping approaches has been investigated in a number of conditions, notably retinopathies [13,14,15,16] and myopathies [10,14]. In particular, a few studies have already reported on the use of dual AAV vectors for DMD therapeutic correction [17,18] and even a triple vector study reported on the ability to reconstitute a full coding sequence for dystrophin by using a trans-splicing vector system, although with a very low efficiency [19]. However, none of these approaches investigated the efficiency of transgene reconstitution determined by homologous recombination efficiency of the overlapping sequences or the formation of aberrant products driven by ITR concatemerization. The quantification of this parameter would impact on the use of a clinical dose as well as on the safety of the system and is therefore necessary for therapeutic translation since these products represent a potential drawback of the dual system for a therapeutic transability.

In our strategy, we aimed at generating a longer form of dystrophin that contained additional domains, as compared to µDys, that have been shown to be required for their structural and signaling roles [20,21,22,23]. To check the suitability of the approach for therapeutic use, we assessed in vivo gene transfer efficacy by employing a ddPCR-based strategic primer design that detects HR-based reconstitution as well as ITR-dependent concatemerization events. Moreover, we evaluated the transgene expression specificity by monitoring any expression activity coming from the single vectors to control the presence of any useless and unwanted peptides in circulation. By using the newly developed methodologies, we showed that our dual AAV system is a candidate for DMD gene therapy as it provided significant therapeutic benefits that were associated with high and specific transgene reconstitution efficiency.

## 2. Results

### 2.1. Design of the Dual Vector System for Quasidystrophin Expression

In order to produce an ectopic dystrophin variant closer to the native protein than uDys, we developed a dual AAV vector system producing a 7.5 kb cDNA called quasidystrophin that preserves four additional key domains as compared to the currently used µDys: (1) the nNOS binding domain (R16–17) [21,24], (2) the Par1b binding domain (R8–R9) [25,26], (3) the R20–23 microtubule binding domain [27] and (4) a full C-terminal domain [20,28,29]. To generate a dystrophin coding sequence including these additional regions, two AAV vectors carrying parts of a codon-optimized transgene, that can reconstitute the full transgene at a DNA level by homologous recombination of the overlapping region were constructed (Figure 1A). The 5′ AAV vector contains the skeletal and cardiac muscle-specific promoter sp512 [30], followed by a chimeric intron and a Kozak sequence to ensure proper initiation of protein translation. The coding sequence includes the N-terminal actin-binding domain, the Hinge 1, R1–R3, Hinge 2, R8–9, R16–17, Hinge 3 and R20. The homologous region, 800 bp in length, was constituted by a portion of R16, Hinge 3 and a portion of R20. The 3′ AAV vector contained the HR region, the completion of R20 until R24, the Hinge 4 and the SV40 polyA signal. Our first aim was to test the therapeutic efficacy of the final product and its suitability for gene therapy. For this purpose, we developed tools to monitor the desired events, namely HR-based reconstitution and specific expression of the full transgene, while also monitoring unwanted events, such as concatemerization or expression of truncated proteins, (Figure 1B).

### 2.2. Systemic Delivery of Dual Aav9 Vectors Leads to Robust Expression of Quasidystrophin in Dba/2j-Mdx Skeletal Muscles

To check the reconstitution of the transgene in vivo, we injected systemically 1-month-old DBA2 mdx mice, an aggressive mdx mouse model, with 5′ and 3′ AAV at a dose of 4 × 10^13^ viral genome (vg)/kg for each vector in order to obtain a total dose similar to or below what has been used in clinical trials [31]. We investigated protein expression levels in the tibialis anterior (TA), gastrocnemius (GA) and diaphragm (DIA), collected 8 weeks post-injection, using immunohistochemistry (IHC) and western blot (Figure 2A–C). Collection time was selected in order to assess both protein expression and eventually therapeutic efficacy at histological and functional level. Dystrophin staining on transversal sections of muscle showed nearly 100% dystrophin-positive fibers in all mdx muscles treated with dual AAVs after quantification of the total number of fibers visualized by laminin staining (Figure 2B). Quasidystrophin was additionally detected by western blot analysis for the appropriate size of the protein coded by our reconstituted transgene (280 kDa), using an antibody recognizing the C-terminal domain of dystrophin (Dys2). Quantification analysis performed by normalizing for GAPDH showed that dystrophin expression was significantly higher than the full-length dystrophin in wt mice, at up to 10-fold in Dia and TA (Figure 2C), taking into consideration the fact that accurate quantification could be affected by transfer efficiency of the full-length dystrophin. Overall, the data indicate that efficient dystrophin restoration is achieved in skeletal muscles upon systemic delivery in mdx mice.

### 2.3. Quantification of Homologous Recombination and Concatemerization Efficiency In Vivo

We developed a method to quantify the efficiency of transgene reconstitution by quantification of HR and concatemerization events occurring through ITRs that could enter in competition and affect the efficiency of transgene reconstitution. We thus strategically designed primers for droplet digital PCR (ddPCR) analysis to quantify HR efficiency and concatemerization events occurring through ITR regions of each vector. For HR quantification, we first evaluated the viral copy number (VCN) of each single vector by ddPCR using specific primers recognizing the unique regions for each vector (Figure 3A,B). We then calculated the percentage of the recombined vector using primers spanning the overlapping region in the 5′ and 3′ vector, thus only amplifying the reconstituted transgene. Multiplex reactions were possible thanks to the use of different probe concentrations for FAM or VIC (Figure 3B). Therefore, the ddPCR reaction mix included either primers for single vectors or for the recombined vectors. A ddPCR on the Titin gene was used to normalize for genomic DNA. In order to remove the head-to-tail concatemers that could not be discriminated by the recombined primers (REC), we pre-digested the DNA with SmaI, that recognizes a restriction site within the ITR sequence. Of note is the fact that the internal ITRs are resolved by elimination during HR so that the recombined molecules will not be cleaved. The identity and specificity of the amplified product by ddPCR was verified by gel electrophoresis (900 bp) and sequence analysis that confirmed the identity of the overlapping region. PCR amplification conditions were optimized to exclude any unspecific products. Higher bands of longer concatemers were not detected, ensuring the correct estimation of the recombination efficiency (Appendix A). We calculated the percentage of HR between the AAV vectors based on the ratio of the positive droplets of 5′ + 3′ vector recombined product with respect to the quantification of the less abundant single vector, since it determines the overall possibilities of recombination (Figure 3A,B). For concatemerization analysis, DNA extracted from muscles was first subjected to enzymatic digestion with the unique cutter AleI (Figure 3C) before amplification to obtain fragment lengths suitable for PCR in cases of formation of long concatemers. We used primers close to each ITR of each vector, primers A and B close to the ITRs near the overlapping region of 5′ and 3′ vectors, and primers C and D at the opposite ends, respectively. We placed primer A after primer B, to amplify all the possibilities of concatemer products occurring via ITRs and exclude the HR event that would be detected if primer A was placed before primer B within the overlapping sequence (Figure 3C). The four primers were incorporated in the same reaction since the mixture gave the same result as adding up the concatemerization events derived from the single reactions containing single combinations of primers (Appendix A) and normalized for genomic DNA. The percentage of concatemers was calculated as the sum of the positive droplets versus total number of positive droplets of 5′ + 3′ vectors.

The data showed that, in all muscles analyzed, the HR efficiency of the two AAVs was around 50% in the TA and GA and around 25% in the DIA and that the concatemerization events were 3–5% (10 to 20-fold lower than HR events), so they should not affect HR efficiency. The VCN analyses showed a difference between 5′ and 3′ vectors in the muscle analyzed, with a higher VCN for the 3′ vector, suggesting either a difference in the cellular acquisition of the virus or in the stability of the vector genome. Although the DIA showed a significant lower reconstitution efficiency, as evaluated by the VCN of the recombined vector and the HR efficiency (Figure 3D,E), this seemed not to affect the transduction and the expression level of quasidystrophin, that was shown to be very high in the diaphragm (Figure 2A,B). Finally, we tested whether the efficiency of recombination was affected by the evolution of the disease, in order to evaluate the feasibility of using such an approach in patients with advanced disease. We calculated VCN and HR efficiency in the TA, GA and DIA muscles of 10-month-old DBA2 mdx treated with 5′ and 3′ dual vector at the same dose used for young mice, at 8 weeks post-injection. As shown, although the VCN of recombined qDys is significantly lower than in young mice (Figure 3F and Appendix A), the efficiency of HR is similar to that observed in young mice (Figure 3F,G). This result indicates that HR efficiency is not a parameter affected in animals with advanced disease and that the dual AAV approach can be used in this context.

### 2.4. Specific Expression of the Dual AAV-qDys and Lack of Expression from the Single Vectors

We then analyzed whether the dual AAV approach exclusively induces expression from the recombined product without expression from the single AAV vectors. To check the expression of the reconstituted transgene occurring by homologous recombination and to exclude the possibility of unwanted expression of truncated transgenes coming from single vectors, DBA2 WT mice were systemically injected with either the single vectors separately (5′ or 3′) or together (5′ + 3′) and PBS injected animals were used as a negative control. Muscles were collected for molecular analysis after 4 weeks. To evaluate the absence of truncated versions of quasidystrophin coming from hypothetical 5′ or 3′ transcripts, we used a double antibody hybridization in Western blot experiments. Practically, we used N-terminal and C-terminal dystrophin antibodies (Figure 4A) that can recognize the protein products that could originate from 5′ and/or 3′ vectors (if any). The WB reveals the exclusive production of quasidystrophin protein when mice are co-injected with 5′ and 3′ dual AAV vectors (Figure 4C). To confirm the absence of expression of aberrant products coming from the single vectors, we also evaluated dystrophin gene expression. For this purpose, we checked mRNA expression by retrotranscription of RNA extracted from muscle and amplification by ddPCR using the primers spanning the overlapping region in the 5′ and 3′ to detect the recombined transcript and primers recognizing uniquely the 5′ or 3′ vectors. We showed specific mRNA expression from muscle of mice co-injected with the 5′ and 3′ vectors, only minor mRNA expression from the 5′ vector equipped with a promoter and no expression from 3′ AAV-treated muscles (Figure 4D). We ruled out any protein expression from the 5′ vector that showed low levels of mRNA by performing dystrophin staining in the TA muscle 1 month after systemic injection of dual AAV-5′ in DBA2-mdx mice, where we could only detect negligible levels of dystrophin staining (Appendix A). Overall, these data show that, once injected in mice, the dual 5′ and 3′ vectors only lead to expression of the recombined transgene and do not express any truncated transgenes from either single vector.

### 2.5. Dual AAV qDys Improves Muscle Pathology and Muscle Function in Dystrophic Mice

We evaluated the therapeutic efficiency of the dual vector AAV approach in DBA2 mdx mice by looking at histological and functional parameters. As shown, GA and DIA muscles of mdx mice treated systemically with the 5′ and 3′ vector, each at the dose of 4 × 10^13^ vg/kg, showed rescue of a healthy histological state with a decrease in the necrotic area (arrows) and centralized nuclei, as visualized by HE, and a reduction in the fibrotic area, visualized by Sirius red coloration (Figure 5A,B). Overall, the histopathological analysis revealed a clear amelioration of the dystrophic phenotype in treated mice, with a strong reduction in fibrosis and centronucleation.

The improvement in muscle pathology was also associated with decreased levels of DMD serum creatine kinase (CK) biomarker (Figure 5C) and higher muscle force, evaluated by escape and grip test (Figure 5D), in mice treated with the dual AAV vector for qDys.

## 3. Discussion

Delivery of miniaturized versions of dystrophin delivered by AAV is currently the most promising approach for DMD. However, results from clinical trials and recent investigations have revealed the therapeutic limitations of this strategy, as a full phenotypic rescue cannot be achieved [32,33]. One of the main reasons that account for the partial efficacy of µDys relies on the lack of functional domains that could impact on the proper dystrophin activity. Here, we employed a dual vector strategy based on HR to deliver quasidystrophin, a larger version of the therapeutic gene, as compared to the currently used µDys approach that has the advantage of carrying additional domains crucial to optimum dystrophin function in vivo, but with potential drawbacks related to the use of two vectors. As such, using two AAVs implies the use of a double of the dose only in cases where the efficiency of transgene reconstitution is 100%. In fact, many unwanted competitor events can affect the degree of recombination, such as formation of concatemers through ITRs of each extremity, which could give rise to long concatemers with different orientations and combinations. We describe a methodology here to control safety and efficiency parameters when employing a dual vector approach to delivering a therapeutic gene larger than 4.5 Kb. Therefore, we developed a ddPCR-based strategy that allowed the quantification of both the efficiency of recombination between the homologous regions of the 2 vectors as well as of concatemerization. With this assay, we proved that transgene reconstitution efficiency varied between 30% and 50% depending on the skeletal muscles. We observed a lower HR efficiency in the diaphragm but that did not affect protein expression levels as compared to the gastrocnemius and tibialis anterior. However, it is interesting to investigate why HR efficiency tends to be inferior and it could indicate an impact of the target tissue on the dual strategy. Another ddPCR strategic primer design was applied to check all possible combinations of concatemers occurring through ITRs and our data showed that, in muscle treated with dual AAV for qDys, only negligible events of concatemerizations were detected.

Very interestingly, our data show that HR efficiency is not affected by the advance of the disease, therefore also supporting the use of the dual system in the context of late stages of the disease. Finally, we checked specificity of the expression of the recombined transgene by looking at both mRNA and protein levels and we could not detect any aberrant product from single vectors. These findings paved the way for the investigation of therapeutic efficacy of our approach in vivo using an aggressive mdx mouse model. Our data showed phenotypic rescue of dystrophic mice assessed by significant fibrosis reduction in the gastrocnemius and diaphragm along with decrease CK levels and increase global muscle force. To assess more deeply muscle quality and quantify muscle and fibroadipose tissue upon in vivo treatment, several clinical imaging techniques based on 3D reconstruction or artificial intelligence predictions could be explored in the future to gain further insight into the beneficial effect of our approach [34,35,36].

## 4. Conclusions

Overall, the proposed methodology has the potential to verify the suitability of a dual vector approach for therapeutic translation, while assessing its therapeutic benefits. By using this approach, we showed that dual AAV for qDys has a clinical translational potential as it improves DMD pathology at a clinical dose, assured by the high efficiency of transgene reconstitution and absence of alternative products.

In conclusion, by using the reported methodology, we demonstrated that our dual-AAV vector approach for qDys restoration is suitable for gene therapy application. This method has the advantage of allowing the control of safety and efficiency parameters while investigating the therapeutic benefits, so it should be applied to any approach using multiple vectors to assess their clinical therapeutic potential.

## 5. Materials and Methods

### 5.1. Quasi-Dystrophin Construct and Production of rAAV

The codon-optimized quasi-dystrophin sequence, ΔR4-R7-ΔR10-R-15, ΔR18–19, was based on the human dystrophin sequence (NM_00406.3), optimized to remove the rare human codons, CpG and possible alternative ORFs. DNA fragments were synthesized using GeneCust and cloned into AAV vectors, thus ensuring that the transgene was split in two halves to respect the maximum capacity of the AAV (around 4.7 kb from ITR to ITR). The dual AAV-5′ vector contains the N-terminal domain, Hinge 1, R1–R3, Hinge 2, R8–9, R16–17, Hinge 3 and R20. The 800 bp homologous region (HR) is constituted by partial R16, Hinge 3 and a partial R20. The expression cassette was under the sp512 muscle-specific promoter, followed by chimeric intron and a strong Kozak sequence. The dual AAV-3′ vector contains the HR region, the completion of R20 until R24, the Hinge 4 and the SV40 polyA signal.

For recombinant AAV production, HEK293-T cells (obtained from Stanford University School of Medicine), cultured in suspension, were transfected with the three plasmids coding for the adenovirus helper proteins, the AAV Rep and Cap proteins, and the ITR-flanked transgene expression cassette. Three days after transfection, cells were harvested, chemically lysed and treated with benzonase (Merck-Millipore, Darmstadt, Germany). After filtration, viral capsids were purified by affinity chromatography, formulated in sterile PBS, and the vector stocks were stored at −80 °C. Titers of AAV vector were determined by using quantitative real-time polymerase chain reaction (qPCR). Viral DNA was extracted using the MagNA Pure 96 DNA and viral NA small volume kit (Roche Diagnostics, Indianapolis, IN, USA) according to the manufacturer’s instructions. PCR was performed in an ABI PRISM 7900 HT Sequence Detector with Absolute ROX mix (Taqman, Thermo Fisher Scientific, Waltham, MA, USA), using ITR-specific primers.

The serotype used in the studies was a newly developed engineered capsid AAVg (manuscript in preparation) (EP2021/074964).

### 5.2. Animal Studies

DBA2 (B6;129S4-DBA2^tm1Cpr^/J, strain #000671) and DBA2-mdx (D2.B10-Dmd^mdx^/J) mice were supplied by the Jackson Laboratory. All animal procedures were approved by the National Ethical Committee, C2EA-51 (Evry, France), and the French Ministry of Research (MESRI) and received a national agreement number (APAFIS #1720 and #19736). Four-week-old Dba2_mdx and Dba2_WT male mice were injected by retro-orbital injection with dual dystrophin AAVg vectors at the dose of 4 × 10^13^ vg/kg for each vector (total dose of 8 × 10^3^ vg/kg). AAV9 injected mice were compared to Dba2_mdx mice injected with PBS (negative control) as well as to the Dba2 WT mice (positive control). Eight weeks post-injection, an Escape test was performed (see below), and muscles and serum collected for molecular and histological analysis.

### 5.3. Viral Copy Number and Homologous Recombination Efficiency by ddPCR

Samples (tissue sections and organs) were prepared using the NucleoMag Pathogen kit for viral and bacterial RNA/DNA (Macherey-Nagel, Hoerdt, France, Ref: 744210.4). After that, DNA was extracted using the KingFisher flex machine (Thermo Fisher). Samples were then quantified by Nanodrop (NanoDrop™ 8000) and stored at −20 °C. For VCN analysis of both single AAV vectors and recombined molecules, multiplex channel reactions were performed with the ddPCR Probe system. Extracted DNA (50 ng) was used with the following PCR amplification conditions: initial denaturation at 95 °C for 10 min, followed by 40 cycles of denaturation at 94 °C for 10 s, annealing at 60 °C for 10 s, and transcript extension at 72 °C for 1 min and 30 s. Following this, fluorescent dye stabilization was performed at 98 °C for 10 min. VCN values were calculated using the ratio with the Titin gene amplification. For this analysis, multiplex channels were analyzed, depending on the type/concentration of probe. Primers and probes that were used are listed in Appendix A. For the multiplex channel reaction dedicated to VCN analysis of the reconstituted vector, gDNA was predigested with SmaI before droplet formation and amplification, to remove the head-to tail event of concatemerization. The analysis was performed using the QuantaSoft™ Analysis Pro 1.0.596 software and efficiency of recombination was calculated as copies of DNA/µL of the recombined region divided by the copies of DNA/µL of the limiting vectors (5′ or 3′).

The identity and specificity of the product amplified by ddPCR was checked by sequencing of the amplicon, run on gel electrophoresis after ddPCR amplification. Amplicon recovery from droplets after PCR was performed according to the Droplet digital Application Guide (BioRad, Hercules, CA, USA) to sequence the amplified product. Both amplified gDNA and mRNA amplicons were sequenced.

### 5.4. Quantification of Concatemerization by ddPCR

For quantification of concatemerization, the EVAGREEN system was used. 125 ng of genomic DNA extracted from muscles was digested with the unique cutter AleI (10 units/µg) and used in the ddPCR reaction under the following amplification conditions: denaturation at 98 °C for 30 s, 40 cycles of denaturation at 98 °C for 10 s, annealing at 63 °C for 15 s and transcript extension at 72 °C for 1 min, followed by fluorescent dye stabilization at 72 °C for 10 min. Quantification of concatemerization was calculated as copies of DNA/µL obtained by amplification with the primers that had been designed (Appendix A) divided by the sum of copies of DNA/µL of 5′ + 3′ vectors.

### 5.5. Gene Expression Analysis of Single and Recombined AAV Expression by ddPCR

For RNA extraction, samples were treated with Nucleozol (Macherey-Nagel) and then homogenized with Fast Prep Bead Mill at 6 m/s for 30 s. After that, samples were centrifuged at 12,000× *g* for 15 min at 4 °C. RNA was extracted from the supernatant using the Ideal-32 machine (Innovative Diagnostics, Grabels, France). After the extraction, samples were treated with DNAse (from TURBO DNAse free kit, Life Technologies, Carlsbad, CA, USA) by incubation at 37 °C for 30 min. RNA was then purified using the resin “DNAse Inactivation Reagent” from the same kit. Finally, samples were then analyzed by Nanodrop (NanoDrop™ 8000) to determine the quantity and the quality. For the reverse transcription of mRNA, RevertAid First Strand kit cDNA Synthesis Kit (H) (Thermo Fisher) was used. For this reaction, both random hexamers and oligodT were used at a ratio of 1:10. For each reaction, 50 ng of cDNA was used in the ddPCR-EVAGREEN system under the following amplification conditions: denaturation at 95 °C for 5 min, 44 cycles of denaturation at 95 °C for 30 s, annealing at 55 °C for 15 s and transcript extension at 72 °C for 1 min, followed by fluorescent dye stabilization at 4 °C for 5 min and 90 °C for 5 min, using HR primers listed in Appendix A and carrying a negative control (minus RT) to check for the presence of genomic DNA.

### 5.6. Western Blot Analysis

Proteins were extracted from muscles in RIPA buffer (Ozyme, Saint-Cyr-l’École, France) containing a protease inhibitor cocktail (Roche) and benzonase (Sigma, St. Louis, MO, USA). Depending on the type of tissue, 100–200 µL of Extraction Buffer were utilized. After this, proteins were extracted by homogenization (Fast Prep, 30 s at 6 m/s) and then centrifuged at 10,000× *g* for 15 min. The supernatant was stored at −20°. The extracted proteins were then quantified using the BCA Protein Assay kit (Fisher bioblock, Illkirch-Graffenstaden, France). Following this, the protein extract was complemented with NuPAGE LDS sample buffer (Life Technologies) and DTT (Sigma). Protein extracts were loaded on a tris-acetate 3–8% gel (Life Technologies) and run for 1 h at 150 V. After the run, the proteins were transferred onto nitrocellulose membrane (Life Technologies) using the iBlot 2 Dry system device (Invitrogen, Waltham, MA, USA). The membrane was then incubated in 50% blocking buffer (Li-Cor, Lincoln, NE, USA) and hybridized with primary antibodies overnight at 4 °C. The second hybridization was performed using secondary antibodies coupled to a fluorescent molecule, directed against the primary antibody. Finally, the membrane was scanned on the Odyssey reader. Protein quantification was performed on ImageJ open-source software by normalization with GAPDH or α-actinin-4.

### 5.7. Histology and Immunohistochemistry

TA and GA muscles were cut into sections 8–10 µm thick (Cryostat Leica, Wetzlar, Germany) and placed on a slide. Slides were dried at room temperature, rehydrated with PBS and fixed with PFA 4% plus triton 0.1% for 5 min and blocked with fetal bovine serum (FBS) 5% and goat serum (GS) 5% (*v*/*v* in PBS) for 45 min. The primary antibodies used were laminin (Z0097, DAKO, Les Ulis, France, dilution 1:100) and dystrophin (NCLDYS2, LEICA, dilution 1:20). After hybridization with secondary antibodies, muscle sections were mounted with DAPI (4’,6-diamidino-2-phenylindole), dihydrochloride fluoromount-G (Southern Biotech, Birmingham, AL, USA) and covered with glass slides.

### 5.8. Creatine Kinase Measurement

CK quantification was performed starting with 10 µL of mouse serum by colorimetric assay with the FUJI DRI-CHEM nx500 system (DMV Imaging, Montanay, France), used to measure creatine phosphokinase concentration.

### 5.9. Muscle Force Evaluation

For the escape test, mice are placed inside 30 cm-long tube and attached to a horizontal tension transducer by their tail. In response to gentle pinching of the tail, the mice tried to escape within the tube. A short peak of force was recorded by the force transducer and, typically, 15 pinches are made, and the top 5 pulling tensions are averaged and divided by the weight of the mouse. The test can be done only once in a lifetime because of a memory effect. Data are reported as maximum peak and the mean of the five peaks normalized to body weight. In the 2-limb grip strength test, the grip force was measured using the grip strength meter (Bioseb https://www.bioseblab.com/ accessed on 1 December 2021; France Grip Test 25N). Three independent measurements were performed, and the mean value of grip strength normalized to weight was calculated.

### 5.10. Statistical Analysis

All data were analyzed using GraphPad Prism 9.5.1 software. Error bars on plots represent the standard error of the mean (SEM). *p*-values were generated by comparison between 2 biological groups with the “Student *t* test one-tailed” function or two-way ANOVA for multiple comparison tests. Results were considered significant at * *p* ≤ 0.05.

## Figures and Tables

**Figure 1 ijms-24-11421-f001:**
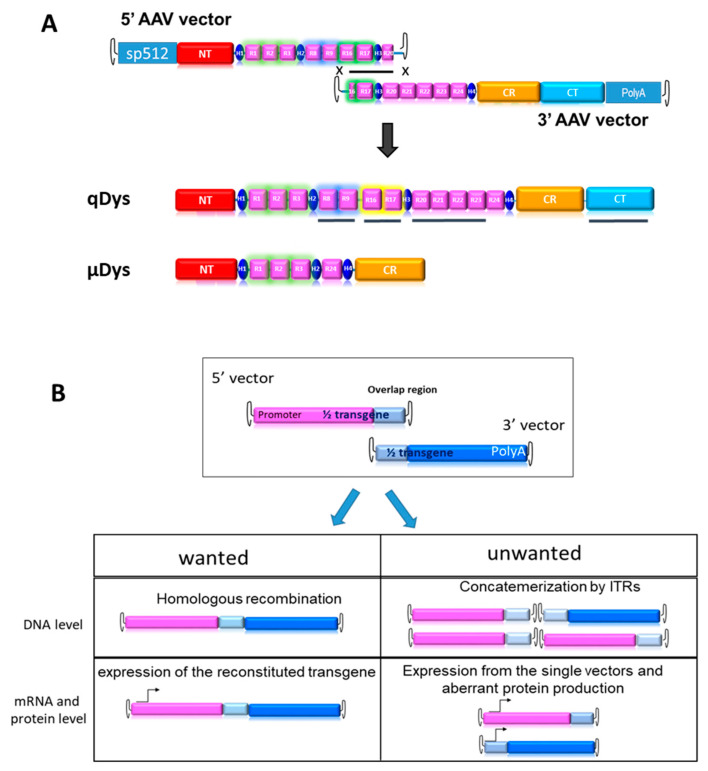
Dual vector strategy. (**A**) HR-driven reconstitution of quasidystrophin (qDys) using 5′ and 3′ AAV vectors. The underlined qDys regions refer to the additional domains present in the final transgene as compared to microdystrophin (µDys) (**B**) Overview of desired and unwanted parameters to check at DNA and expression levels when using the dual AAV technology.

**Figure 2 ijms-24-11421-f002:**
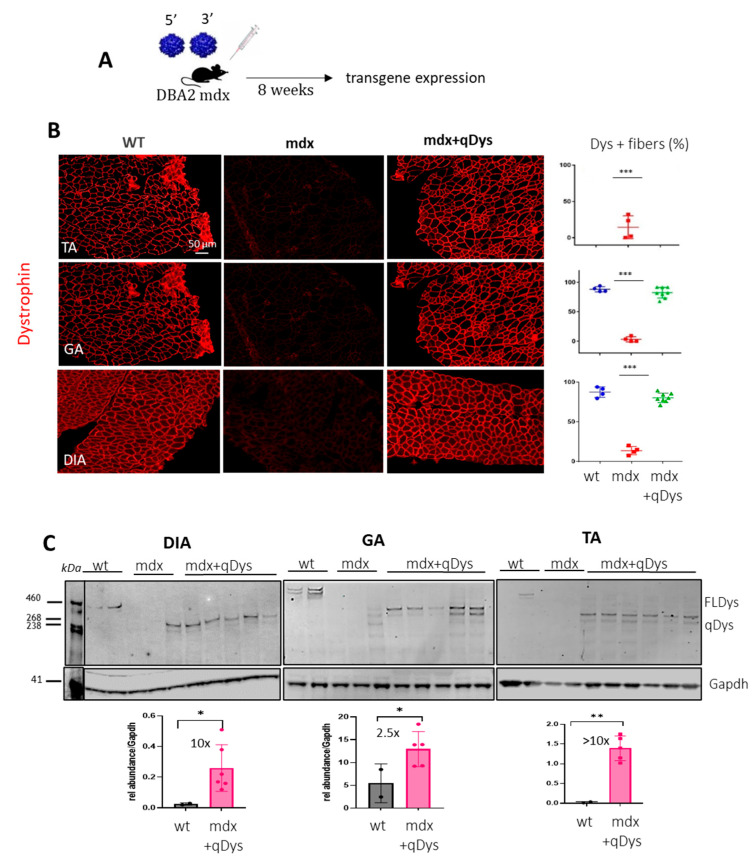
Dual AAV qDys efficiently transduces skeletal muscles. (**A**) Efficiency of muscle transduction was evaluated 8 weeks after systemic delivery of AAV 5′ and AAV 3′ (4 × 10^13^ vg/kg each) in 4-week-old DBA2-mdx. (**B**) Representative images produced from sections of the tibialis anterior (TA), gastrocnemius (GA) and diaphragm (DIA) stained for dystrophin (DysB antibody) and quantification of dystrophin-positive fibers (Dys+) relative to the total number of fibers visualized by laminin staining (in the graph: blue: wt group, red: mdx group, green: mdx treated with qDys group). (**C**) Dual-vector mediated qDys protein expression in TA, GA and DIA, evaluated by WB from total protein lysates in wt, mdx and mdx treated with 5′ and 3′ vectors. Quantification of dystrophin expression levels was performed using the ratio to GAPDH protein for normalization, with ImageJ 1.54d open source software. Statistical analysis was performed with the “Student *t* test one-tailed” GraphPad Prism 9.5.1 software. Data are presented as mean ± SEM (n = 3–5). (* *p* ≤ 0.05, ** *p* ≤ 0.01, *** *p* ≤ 0.001).

**Figure 3 ijms-24-11421-f003:**
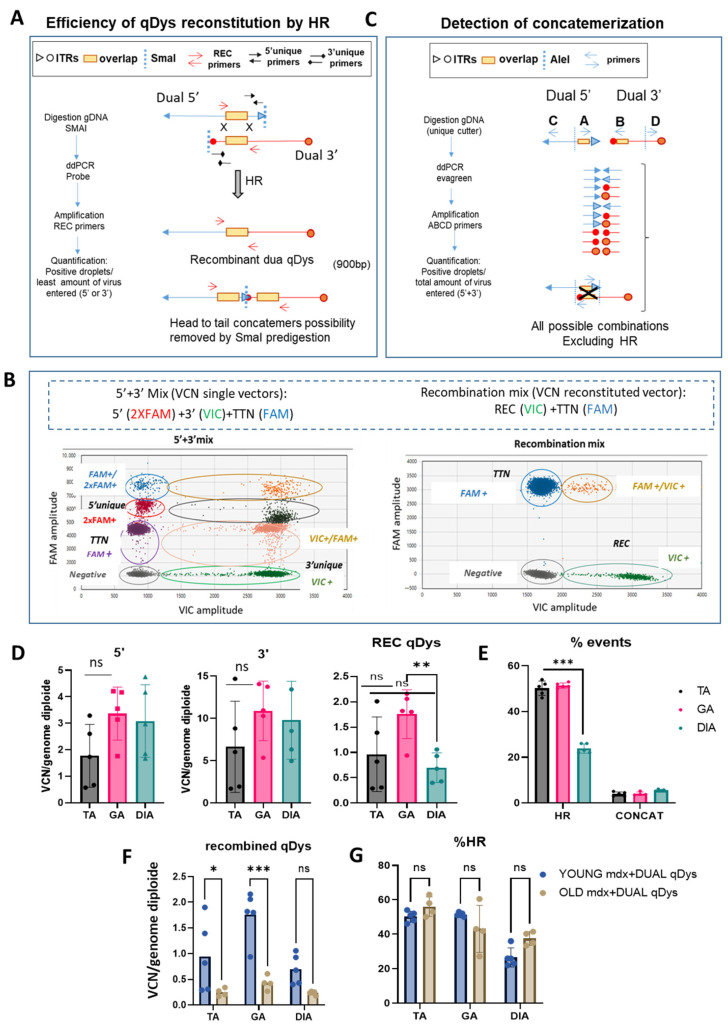
Quantification of HR-dependent reconstitution efficiency of qDys and concatemerization events by ddPCR. (**A**) Schematic representation of the protocol used to calculate HR efficiency using specific primers for VCN analysis of the single and recombined vectors. (**B**) Cloud separation showing single and double positive droplets according to the dye used (VIC or FAM) and the probe concentration (1× or 2×). (**C**) Primer design strategy and protocol to quantify all the ITR-dependent concatemerization events. (**D**) Viral copy number (VCN) of 5′or 3′ vectors and the recombined qDys vector, performed on TA, GA, and DIA of mdx systemically injected with dual vector AAVs. (**E**) Homologous recombination (HR) and concatemerization efficiency, calculated as described in panel A and C and expressed as a percentage. (**F**,**G**) VCN analysis of qDys (**F**) and HR efficiency (**G**) in Young and Old mdx mice co-injected with 5′ and 3′ AAV vectors. Data are presented as mean ± SEM (n = 5) (* *p* ≤ 0.05, ** *p* ≤ 0.01, *** *p* ≤ 0.001). ns, not significant.

**Figure 4 ijms-24-11421-f004:**
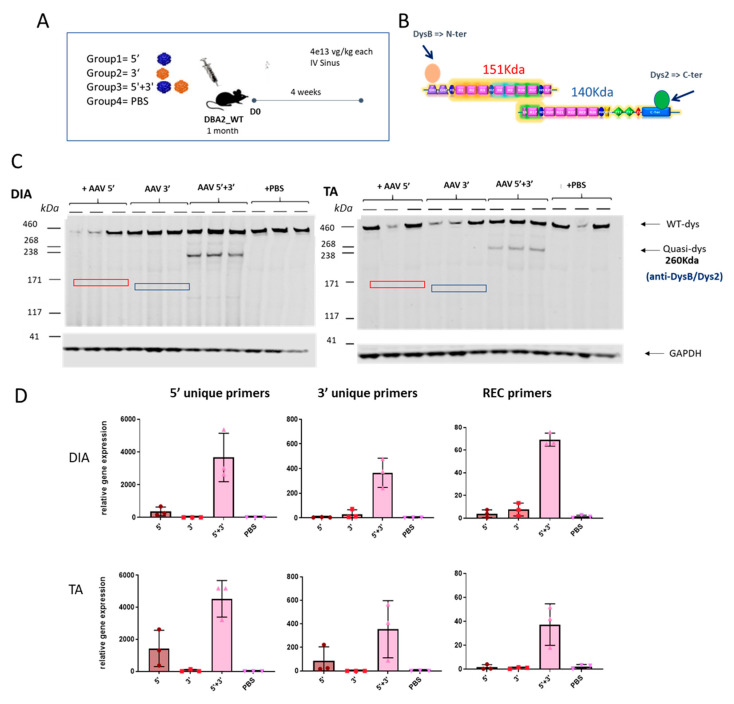
The dual vector system leads to specific expression of the recombined transgene. (**A**) Schematic representation of the experimental animal setting to assess the specific expression of the reconstituted transgene and absence of truncated aberrant dystrophin isoforms coming from the single AAV vectors. (**B**) Scheme of the epitopes recognized by the two dystrophin antibodies used. Dys B antibody recognizes the N terminal part of dystrophin, while Dys 2 recognizes the C terminal part. (**C**) Dystrophin Western Blot results on TA and DIA of mice injected with 5′ and 3′ vectors singly or together and with PBS as a control. Dystrophin protein levels are normalized for GAPDH expression. The hypothetical truncated protein product originating from the 5′ vector would have a molecular weight around 151 kDa (red square), while the hypothetical protein originating from the 3′ vector would have a molecular weight of 140 kDa (blue square). (**D**) RTqPCR analysis performed on RNA extracted from TA and DIA muscles using primers to specifically only the 5′ or 3′ transgene and the recombined transgene (different colors are associated with different groups).

**Figure 5 ijms-24-11421-f005:**
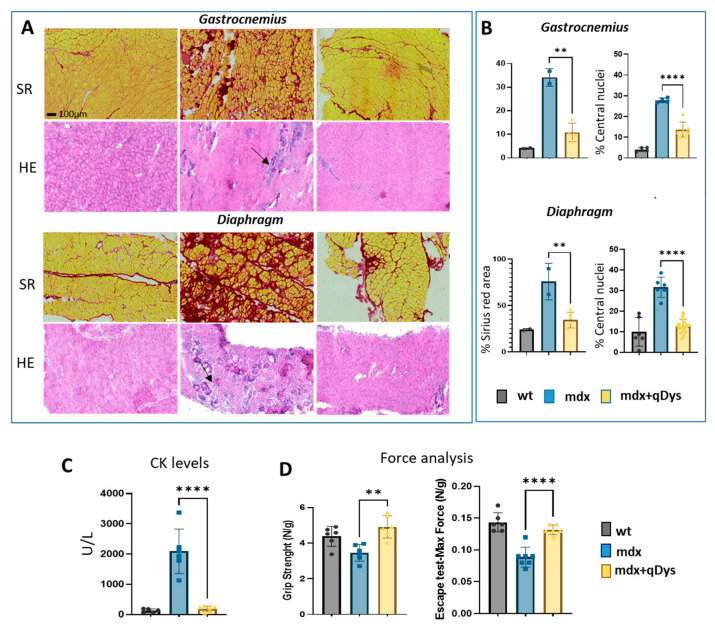
Systemic delivery of dual AAV qDys in mdx mice improves muscle pathology and muscle function. (**A**) Representative sirius red and hematoxilin and eosin staining performed on GA and DIA sections of DBA2 wt, DBA2 mdx and DBA2 mdx that had been systemically treated with dual vector for 8 weeks. Arrows indicate inflammatory and necrotic area. (**B**) Sirius red analysis was performed using QuPath-0.3.2 software and Centro-nucleation analysis was performed with ImageJ open-source software and R studio. Centronucleation is expressed as the percentage of myofibers with centrally located nuclei relative to total number of fibers. (**C**) Creatine kinase levels (U/l) calculated from serum collected 8 weeks after AAV injection. (**D**) Grip test (performed 2 days before euthanasia) and Escape test (day of sacrifice) performed in wt, mdx and mdx where the dual vector was delivered (mdx + qDys). Force values were normalized for body weight (N/g). Data are presented as mean ± SEM (n = 6) (** *p* ≤ 0.01, **** *p* ≤ 0.001).

## Data Availability

Not applicable.

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
