# Peer review of "Assessment of Therapeutic Potential of a Dual AAV Approach for Duchenne Muscular Dystrophy"

_ijms, 2023, doi:10.3390/ijms241411421_

Round 1

Reviewer 1 Report

The abstract is a honest summary of a well designed and implemented study. I have no major concerns, but for the general readers I suggest to add and discuss a few more References on Clinical Imaging that may improved the study and the paper, specifically:

Recenti M, Ricciardi C, Edmunds K, Jacob D, Gambacorta M, Gargiulo P. Testing soft tissue radiodensity parameters interplay with age and self-reported physical activity. Eur J Transl Myol. 2021 Jul 12. doi: 10.4081/ejtm.2021.9929.

Recenti M, Ricciardi C, Edmunds K, Gislason MK, Gargiulo P. Machine learning predictive system based upon radiodensitometric distributions from mid-thigh CT images. Eur J Transl Myol. 2020 Apr 1;30(1):8892. doi: 10.4081/ejtm.2019.8892. eCollection 2020 Apr 7

Gargiulo P, Helgason T, Ramon C, Jónsson H Jr, Carraro U. CT and MRI Assessment and Characterization Using Segmentation and 3D Modeling Techniques: Applications to Muscle, Bone and Brain. Eur J Transl Myol. 2014 Mar 27;24(1):3298. doi: 10.4081/ejtm.2014.3298. eCollection 2014 Mar 31.

No comments

Author Response

Minor points:

The abstract is a honest summary of a well designed and implemented study. I have no major concerns, but for the general readers I suggest to add and discuss a few more References on Clinical Imaging that may improve the study and the paper,

RE: Thanks for the suggestion, we included the 3 references in the discussion of the manuscript

(page 11)

 Minor editing of English language required

RE: We also re-edited the manuscript for English language as suggested

Reviewer 2 Report

The authors conducted a very important study to assess dual AAV approach and its efficiency of transgene reconstitution in dystrophic mice. This study is timely needed.

1. Line 66 - 'and myopathies' - font/size should be rectified.

2.  Can you explain why the bands were expressed at 268 kDa in western blot results 2C?

3. Why 8 weeks after systemic delivery of AAVs was selected to evaluate the efficiency of muscle transduction?

4.  Discussion part could have been expanded further with appropriate citations.

Author Response

Minor points:

  1. Line 66 - 'and myopathies' - font/size should be rectified.

RE: thanks, it has been rectified

  1. Can you explain why the bands were expressed at 268 kDa in western blot results 2C?

RE: The size of the protein coded by the reconstituted transgene, after homologous recombination of 5’ and 3’ is 280kDa. That is better explained in line 126 (highlighted in yellow)

  1. Why 8 weeks after systemic delivery of AAVs was selected to evaluate the efficiency of muscle transduction?

RE: This is a good point that we clarified in line 122. It is true that 3-4 weeks would be sufficient to monitor gene transfer efficiency. However, our goal was to test both gene transfer and therapeutic efficacy at histological and functional levels which requires longer time. Therefore, we selected 8 weeks post-injection to be able to assess a longer-term stable expression and to assess the efficacy of parameters like fibrosis and muscle function. We explained that in line 122/123, highlighted in yellow

  1. Discussion part could have been expanded further with appropriate citations.

RE: We extended the discussion and included appropriate references, as indicated in the yellow (page 11).